# Preliminary efficacy of cognitive-behavioral therapy on emotion regulation in adults with autism spectrum disorder: A pilot randomized waitlist-controlled study

Miho Kuroda[1,2]*, Yuki Kawakubo[1]*, Yoko Kamio[3], Hidenori Yamasue[4], Toshiaki Kono[5], Maiko Nonaka[1], Natsumi Matsuda[1], Muneko Kataoka[1], Akio Wakabayashi[6], Kazuhito Yokoyama[7], Yukiko Kano[1], Hitoshi Kuwabara [1,4,8]*

1 Department of Child Neuropsychiatry, Graduate School of Medicine, The University of Tokyo, Bunkyo-ku, Tokyo, Japan, 2 Department of Psychology, Faculty of Liberal Arts, Teikyo University, Hachioji, Tokyo, Japan, 3 Department of Child and Adolescence Mental Health, National Institute of Mental Health, National Center of Neurology and Psychiatry, Kodaira, Tokyo, Japan, 4 Department of Psychiatry, Hamamatsu University School of Medicine, Hamamatsu, Shizuoka, Japan, 5 Department of Community Mental Health & Law, National Institute of Mental Health, National Center of Neurology and Psychiatry, Kodaira, Tokyo, Japan, 6 Department of Psychology, Doctoral Program of Human Information Sciences, Graduate School of Advanced Integrated Sciences, Chiba University, Chiba, Chiba Prefecture, Japan, 7 Department of Epidemiology and Environmental Health, Faculty of Medicine, Juntendo University, Bunkyo-ku, Tokyo, Japan, 8 Department of Psychiatry, Saitama Medical University, Moroyama-chou, Saitama, Japan

* kuwabah-tky@umin.ac.jp (HK); mihok-tky@umin.ac.jp (MK); kawakuboyuki@gmail.com (YK)

**Data Availability Statement:** All relevant data are within the paper and its Supporting information files.

## Abstract

Previous studies have demonstrated the clinical utility of cognitive-behavioral therapy in improving emotion regulation in children on the autism spectrum. However, no studies have elucidated the clinical utility of cognitive-behavioral therapy in improving emotion regulation in autistic adults. The aim of the present pilot study was to explore the preliminary clinical utility of a group-based cognitive-behavioral therapy program designed to address emotion regulation skills in autistic adults. We conducted a clinical trial based on a previously reported protocol; 31 participants were randomly allocated to the intervention group and 29 to the waitlist control group. The intervention group underwent an 8-week program of cognitive-behavioral therapy sessions. Two participants from the intervention group withdrew from the study, leaving 29 participants (93.5%) in the group. Compared with the waitlist group, the cognitive-behavioral therapy group exhibited significantly greater pre-to-post (Week 0–8) intervention score improvements on the attitude scale of the autism spectrum disorder knowledge and attitude quiz ($t = 2.21$, $p = 0.03$, $d = 0.59$) and the difficulty describing feelings scale of the 20-item Toronto Alexithymia Scale ($t = -2.07$, $p = 0.04$, $d = -0.57$) in addition to pre-to-follow-up (Week 0–16) score improvements on the emotion-oriented scale of the Coping Inventory for Stressful Situations ($t = -2.14$, $p = 0.04$, $d = -0.59$). Our study thus provides preliminary evidence of the efficacy of the group-based cognitive-behavioral therapy program on emotion regulation in autistic adults, thereby supporting further evaluation of the effectiveness of the cognitive-behavioral therapy program in the context of a larger randomized clinical trial. However, the modest and inconsistent effects underscore

**Funding:** This study was supported by an Intramural Research Grant (23–1) from the Neurological and Psychiatric Disorders program of the National Center of Neurology and Psychiatry and Grants-in-Aid (No. 23119706 and No. 22531078). The funders had no role in study design, data collection and analysis, decision to publish, or preparation of the manuscript.

**Competing interests:** The Authors have declare that no competing interests exist.

the importance of continued efforts to improve the cognitive-behavioral therapy program beyond current standards.

## Introduction

Adults with autism spectrum disorder (ASD) frequently experience mental health problems, in addition to social communication impairments and repetitive and restricted behaviors as their core symptoms. Anxiety and depression are the most common comorbid symptoms of ASD [1–4]. Higher rates of anxiety and depression in autistic individuals have been associated with lower life satisfaction and greater social difficulties [5, 6]. However, the mechanisms underlying these high rates of comorbid anxiety and depression remain speculative. Studies have begun to focus on the potential importance of emotion regulation as a mental health risk factor in ASD [7–9].

Emotion regulation is generally defined as the automatic or intentional monitoring and modification of a person's emotional state that promotes goal-directed behavior [10, 11]. Studies on emotion regulation in both autistic children and adults have primarily focused on differences in the implementation of a particular emotion regulation strategy, in addition to the relationship between particular emotion regulation and a range of outcomes, including mental health and social functioning [12, 13]. These studies have shown that, compared with typically developing (TD) participants, autistic individuals implement maladaptive emotion regulation strategies, such as emotion-oriented strategies, rather than adaptive emotion regulation strategies, such as task-oriented strategies [9, 14].

An important research focus linked to emotion regulation in ASD is to identify factors that lead to emotion regulation difficulties. Among the many frameworks for conceptualizing emotion regulation, the process model, which describes emotion regulation as a multicomponent and dynamic process between the individual and their context, is the most influential and widely known. This model distinguishes three stages of emotion regulation: identification (whether to regulate or not), selection (deciding which strategy to select), and implementation (implementing a strategy) [15]. In the identification stage, the process model emphasizes the importance of alexithymia, which refers to difficulty in identifying and labeling one's own emotions; empirical studies have indicated increased levels of alexithymia in autistic individuals compared with TD individuals [9, 16]. With regard to the selection stage, the process model suggests that emotion regulation involves adopting an alternative point of view on emotional appraisals; these strategies require abilities which are impaired in ASD, such as theory of mind [15, 17]. Previous studies have suggested that theory of mind, i.e., the ability to attribute mental states to others to make sense of their behavior, is atypical in ASD [18, 19]. However, it remains unknown whether a universal pattern of cognitive impairment in ASD exists and whether multiple cognitive impairments are needed to explain its full range of behavioral symptoms [20]. Social cognition clearly encompasses a range of processes, including, but not limited to, theory of mind and emotion processing, which appear to be distinct but interdependent [21]. Studies of autistic individuals as well as studies of TD individuals both suggest that the skills involved in comprehending the self and others are interrelated and play an important role in emotion regulation [14, 22].

Previous studies have demonstrated the effectiveness of cognitive-behavioral therapy (CBT) in improving anxiety [23–26], anger [27], and emotion regulation [28–30] in autistic children and adolescents. However, few studies have investigated its efficacy in autistic adults and only

one randomized controlled trial has assessed the effectiveness of CBT in alleviating obsessive-compulsive disorder (OCD) symptoms in autistic adults [31]. To our knowledge, no studies have elucidated the effectiveness of CBT in improving emotion regulation in autistic adults.

Usually, parents are involved in their child's CBT sessions [29, 30]. As such, parents gain a greater understanding of ASD and help guide emotion regulation training in light of ASD traits. However, unlike in children, autistic adults need to better understand their own strengths and weaknesses, as they are expected to modify their own emotional regulation skills without their parents' help. In addition, an ASD diagnosis can have adverse effects on emotions, including stigmatization and lowered self-esteem; however, having accurate knowledge of and a positive attitude towards an ASD diagnosis can foster self-awareness in a way that is matched with minimal self-criticism [32].

The aim of the present pilot study was to explore the preliminary efficacy of CBT on emotion regulation in autistic adults [33]. To this end, we conducted a group-based CBT program for emotion regulation in conjunction with psychoeducation about ASD to help autistic adults (1) acquire effective emotion regulation strategies, (2) increase their own emotional awareness, (3) improve their capacity to comprehend others' emotions, and (4) increase their knowledge and improve their attitudes concerning ASD.

## Materials and methods

### Study design

We conducted a randomized controlled trial based on a previously reported protocol [33]. We used a single-blinded trial design in which the assessors did not attend the intervention sessions and did not know of the participants' group assignments. We used a waitlist (WL) control group consisting of autistic individuals who received their usual services. After the study, participants in the WL control group had the option to receive the CBT program.

The trial was registered in the University Hospital Medical Information Network Clinical Trials Registry and was approved by the International Committee of Medical Journal Editors (No. UMIN000006236).

### Ethical considerations

The study design was reviewed and approved by the institutional review board of The University of Tokyo Hospital (No. 2702) and the National Institute of Mental Health (No. A2010-022). All procedures were performed in accordance with the principles of the Declaration of Helsinki. All participants provided written informed consent after receiving a complete explanation of the trial.

### Participants and randomization

Participants were individuals diagnosed with ASD. They were recruited through pools of research volunteers at two sites, the Department of Child Psychiatry, University of Tokyo Hospital (Site 1) and the Department of Child and Adolescent Mental Health, National Institute of Mental Health in Tokyo (Site 2). Recruitment information was advertised on the University of Tokyo Hospital's website.

Participants were required to meet six inclusion criteria. (1) Participants were required to have a confirmed diagnosis of a pervasive developmental disorder (i.e., autistic disorder, Asperger's disorder, or pervasive developmental disorder not otherwise specified) according to the Diagnostic and Statistical Manual of Mental Disorders, Fourth Edition, Text Revision (DSM-IV-TR) [34]. This diagnostic confirmation was to be based on the Autism Diagnostic

Schedule (ADOS) [35]; the Autism Diagnostic Interview, Revised (ADI-R) [36]; the Autism-Spectrum Quotient (AQ), Japanese version [37]; the Social Responsiveness Scale for Adults, Japanese version (SRS-A) [38]; the Social Communication Questionnaire, Japanese version (SCQ) [39]; or the Empathizing-Systemizing Quotient, Japanese version (ESQ) [40]. (2) Participants were required to be between 18 and 50 years old. (3) Participants were required to have a full intelligence quotient (FIQ) of at least 85 and a verbal intelligence quotient (VIQ) of at least 100, as assessed by the Wechsler Adult Intelligence Scale, Third Edition [41]. (4) Participants had to have graduated from high school. (5) Participants had to be aware of their ASD diagnosis. (6) Participants needed to have an awareness of their lack of emotional self-awareness and their poor comprehension of the emotions of others. Participants' awareness of their lack of emotional self-awareness and poor comprehension of others' emotions was confirmed through a direct interview with a PhD-qualified psychologist. The exclusion criterion was unstable comorbid mental disorder symptoms evaluated using the Mini-International Neuropsychiatric Interview [42] and diagnosed according to the DSM-IV-TR [34].

Participants were randomly allocated to groups through the University Hospital Clinical Trial Alliance Clinical Research Supporting System (UHCT ACReSS) with sex as the allocation factor. To achieve blinded allocations, the UHCT ACReSS provided a random allocation sequence such that the researchers and staff were unaware of it until after the enrollment period.

The pre-intervention assessments were completed at sites 1 and 2. The post-intervention and follow-up (FU) assessments were completed at Site 1. Blinded assessors conducted pre-intervention assessments within four weeks of the start of the interventions (Week 0) and post-intervention assessments within four weeks from the end of the interventions (Week 8), and FU assessments 8–12 weeks later (Week 16) (Fig 1).

We used G*Power statistical software [43] to estimate the target number of participants. The CBT program for emotion regulation in autistic children [30] had a significant clinical impact and hence powered this study to detect a large effect (Cohen's d = 0.80). A significance level ($p$-value) of 0.05 (two-tailed), a test power of 0.8, and a layout proportion of 1 yielded an estimated requirement of 52 participants. Therefore, we used a sample size of 60 to account for potential dropouts.

## Intervention

The CBT program we implemented was a group-based intervention that targeted emotion regulation improvements in autistic adults. Each CBT session lasted approximately 100 minutes, with weekly sessions that took place over an eight-week period. If a participant could not attend a group session, a supplementary individual session was offered up to three times. Two therapists and 4–5 participants formed each CBT group. The group leader was a certified, PhD-qualified psychologist with over 10 years' experience assisting ASD clients. The sub-leader was a psychologist with a master's degree.

We developed manuals for the CBT program on emotion regulation and psychoeducation about ASD for autistic adults with references to previous CBT studies. Each CBT session included a psychoeducation part on ASD (except for sessions 2, 7, and 8, which were dedicated to emotion regulation only) and an emotion regulation part (except for session 1, which was dedicated to psychoeducation on ASD). There was a short relaxation period between the two parts in each session (Table 1).

The materials for the ASD psychoeducation, including an original handout, were developed to help participants learn about and understand the nature of ASD. During the psychoeducation part, the group leader presented a lecture about diagnostic criteria, symptoms, etiology,

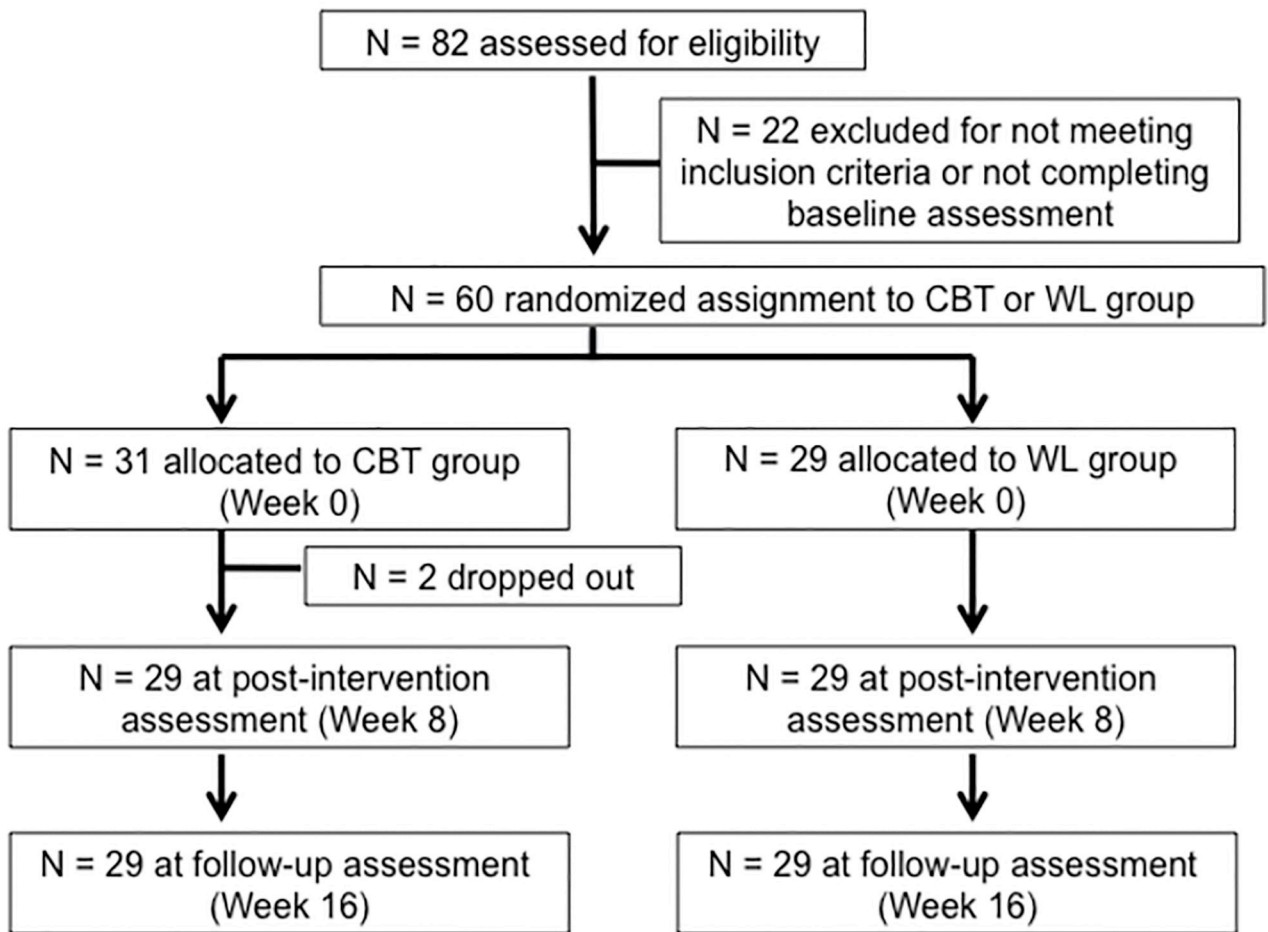

**Fig 1. CONSORT flow diagram.** Similar numbers of participants were allocated to the CBT and WL groups. Assessments were performed within four weeks prior to the start of the CBT/waiting period (pre-intervention), within four weeks after the conclusion of CBT/waiting period (post-intervention), and 12 weeks thereafter (follow-up). Abbreviations: CBT, cognitive-behavioral therapy; WL, waitlist.

and daily life. Participants were asked to explain their preferences, strengths, and weaknesses verbally, and a group discussion was held.

In the emotion regulation part, the interventions started with emotion education (such as recognizing and labeling emotions in self and others) in a concrete manner such that ASD participants could comprehend and externalize their emotions. The Cat-kit Japanese version [44] was used for the emotion regulation part of the CBT program to complement the weaknesses inherent to ASD. Based on randomized controlled trials which investigated CBT's efficacy in improving emotional symptoms in autistic children and adolescents [23, 27], the Cat-kit was developed [45] to help autistic individuals become aware of how their thoughts, feelings, and actions interact, and how to use various visual tools to share their insights with others. Subsequently, the sessions progressed to cognitive reconstructions using more complex skills, based on relaxation and/or planned systematic exposure. Finally, problem solving skills, such as assertion skills and coping skills, were explained and discussed with the group. These interventions were applied across multiple emotions with a focus on learning and practicing various adaptive emotion regulation strategies.

**Table 1. Overview of the CBT program.**

| Duration | Content |
|---|---|
| 5 min | Greeting and checking current emotion |
| 30 min | Session 1. Diagnosis and epidemiology of ASD<br>Session 2. How to relax<br>Session 3. Making a book of my favorite things<br>Session 4. My unique points compared with others<br>Session 5. My strengths<br>Session 6. My weakness and how to deal with them<br>Session 7. Study of anger<br>Session 8. Assertion |
| 10 min | Relaxation (deep breathing and light physical exercises) |
| 40 min | Session 1. Characterization of ASD<br>Session 2. Joy<br>Session 3. Safety<br>Session 4. Affection<br>Session 5. Anxiety<br>Session 6. How to reduce anxiety<br>Session 7. How to reduce anger<br>Session 8. Coping skill |
| 5 min | Rest period |

Abbreviations: CBT, cognitive-behavioral therapy; ASD, autism spectrum disorder.

## Outcome measures

**Primary outcomes.**   To determine the direct effects of the CBT intervention, we assessed the effects of changed intervention periods on the scores of the Coping Inventory for Stressful Situations, Japanese version (CISS) [46], the 20-item Toronto Alexithymia Scale Japanese version (TAS20) [47], the Motion Picture Mind-Reading task (MPMR) [48], and the ASD knowledge and attitude quiz (ASD-Q) (S2 File).

The CISS, a 48-item self-reported questionnaire, is used to determine an individual's preferred emotion regulation strategies. It features subscales for task-oriented (CISS-T), emotion-oriented (CISS-E), and avoidance-oriented (CISS-A) coping styles. CISS-T assesses emotion regulation strategies such as cognitive reappraisal, CISS-E assesses emotion regulation strategies such as self-blaming, and CISS-A assesses emotion regulation strategies such as engaging in alternative activities. Each item is rated from 1 ("not at all") to 5 ("very much"), and total scores for each subscale range from 16 to 80. Higher scores on the CISS-T indicate more adaptive strategies with sufficient emotion regulation. Higher scores on the CISS-E indicate more maladaptive strategies with difficulties in emotion regulation. Higher scores on the CISS-A indicate both adaptive and maladaptive strategies [9, 49].

The TAS20, which assesses own emotional awareness, is frequently used to measure alexithymia. Individuals with alexithymia have trouble identifying and describing their emotions, and tend to focus attention outside themselves. This self-reported assessment features seven questions about difficulties in identifying feelings (TAS20-F1), five questions about difficulties in describing feelings (TAS20-F2), and eight questions about externally oriented thinking (TAS20-F3). Each item is answered on a 5-point Likert-type scale. Total scores (TAS20-total) range from 20 to 100, and higher scores indicate more severe alexithymia.

The MPMR [48] assesses the capacity to comprehend others' emotions and tests "theory of mind" abilities by means of a task that requires participants to assess whether certain emotion words suitably describe people in video clips from a television drama (*Shiroi Kyotō*). This task

includes 41 video clips, each lasting 3–11 seconds, with concurrent auditory and visual presentations. The correct response percentage represents the task performance.

The ASD-Q is a self-report questionnaire developed specifically for this study that was used to assess knowledge and attitudes regarding ASD [33]. The ASD-Q comprises ten knowledge-based questions and five attitude-based questions about ASD. Each knowledge-based question has three response options ("true," "false," or "do not know"). The score is calculated according to the number of correct answers. The highest possible score is 10, and higher scores indicate greater knowledge of ASD. Each attitude-based question is answered on a 5-point Likert-type scale. The highest possible score is 25, and higher scores indicate a more positive attitude towards ASD.

**Secondary outcomes.** Secondary outcome measures included scores from the CISS, TAS20, MPMR, and ASD-Q at the 12-week FU assessment.

We speculated that participants would show greater adaptation and experience an improvement in anxiety and depressive symptoms due to their improved emotion regulation. Therefore, secondary outcomes included scores on the Global Assessment of Functioning (GAF), the 26-item World Health Organization Quality of Life scale (QOL) [50], the Liebowitz Social Anxiety Scale (LSAS) [51], the State-Trait Anxiety Inventory (STAI) [52], the Social Phobia and Anxiety Inventory (SPAI) [53], and the Center for Epidemiologic Studies Depression Scale (CES-D) [54]. The GAF and LSAS are clinician-reported measures, and the others are self-reported measures. Given the trial's single-blinded nature, raters who were blinded to group assignments conducted the clinician-reported assessments.

## Statistical analyses

The analyses were performed according to the intention-to-treat (ITT) principle. The ITT analysis of the results in this study was based on the initial treatment assignment, but not on the treatment eventually received.

Changes between pre-intervention and the post-intervention assessments were herein referred to as intervention period changes, and changes between pre-intervention and FU assessments were referred to as study period changes. We used independent *t*-tests (two-tailed) to compare the intervention or study period score changes between the CBT and WL groups. We used a *p* value of 0.05 as the significance threshold. The effect sizes were assessed using Cohen's *d*. Statistical analyses were performed using SPSS 20 J (IBM, Armonk, NY).

## Results

### Recruitment and participant flow

Participant registration began on September 1, 2011, and the FU period ended on June 22, 2013.

We recruited 82 autistic individuals and assessed them based on the inclusion criteria. Of the 82, 60 (73.2%) met the inclusion criteria. We obtained informed consent and then enrolled the eligible participants in the trial through the UHCT ACReSS at the University of Tokyo.

Of the 60 included participants, 29 (21 men and eight women; mean age: 29.6 ± 8.0 years) and 31 (20 men and 11 women; mean age: 32.7 ± 8.1 years) were allocated to the WL control group and the CBT group, respectively. Two participants in the CBT group (one man and one woman) withdrew from the study (the woman was too busy, and the man lost his motivation to continue), yielding a final total of 29 participants (19 men and 10 women; mean age: 29.6 ± 8.0 years) in the CBT group (Fig 1).

**Table 2. Descriptive statistics of participant characteristics.**

| | | CBT | WL |
|---|---|---|---|
| N | | 31 | 29 |
| Male/female | | 20/11 | 21/8 |
| DSM-IV subtype: Au/Asp/Nos | | 20/2/9 | 20/2/7 |
| Comorbid disorder/no comorbid disorder | | 18/13 | 14/15 |
| Medication/no medication | | 21/10 | 18/11 |
| | | Mean (SD) | Mean (SD) |
| Age, years | | 32.7 (8.1) | 29.6 (8.0) |
| IQ | Full | 110.2 (12.9) | 104.9 (11.9) |
| | Verbal | 114.8 (14.7) | 109.2 (9.6) |
| | Performance | 102.6 (15.1) | 98.0 (15.4) |
| ADOS | Communication | 3.2 (1.3) | 3.4 (1.2) |
| | Social interaction | 7.0 (1.6) | 7.3 (1.9) |
| ADI-R[†] | Social interaction | 12.0 (6.2) | 12.2 (5.7) |
| | Communication | 9.7 (5.0) | 8.8 (4.2) |
| | Restricted interest | 3.3 (1.9) | 3.3 (1.7) |

[†]Respondents were parents. CBT group, N = 23, WL group, N = 22.

Abbreviations: CBT, cognitive-behavioral therapy; WL, waitlist; DSM-IV, Diagnostic and Statistical Manual of Mental Disorders, Fourth Edition; Au, autistic disorder; Asp, Asperger's disorder; Nos, pervasive developmental disorder not otherwise specified; IQ, intelligence quotient; ADOS, Autism Diagnostic Schedule; ADI-R, Autism Diagnostic Interview, Revised.

The mean attendance rates in the 31 CBT participants were 80.5% for the group sessions alone and 92.5% when additional individual sessions were counted. Eight participants had a 100% group session attendance rate.

## Baseline comparability of characteristics

Table 2 demonstrates that the CBT and WL groups were matched on a range of demographic variables (Table 2).

## Primary outcomes

The results for all outcome measures are presented in Table 3. We found no significant differences in pre-intervention assessments in the CISS-T and CISS-E scores, ASD-Q knowledge scores, TAS20-F1, TAS20-F2, TAS20-F3, and TAS20 total scores, and MPMR scores between the CBT and WL groups. The CISS-A score was the only exception.

The independent $t$-tests showed significant between-group differences in intervention period changes in the TAS20-F2 scores ($t$ = -2.07, $p$ = 0.04, $d$ = -0.57) and the ASD-Q attitude scores ($t$ = 2.21, $p$ = 0.03, $d$ = 0.59). This being said, these findings did not withstand the Bonferroni correction for multiple comparisons for the 10 primary outcome measures ($p < 0.05/10$). We found no significant between-group differences in intervention period changes in the CISS-T, CISS-E, and CISS-A scores, the ASD-Q knowledge scores, the TAS20-F1, TAS20-F3, and TAS20 total scores, and the MPMR scores.

**Table 3. Comparisons of outcome measures for CBT and waitlist groups at pre-intervention (pre), post-intervention (post), and follow-up (FU).**

| Outcome measures | | | CBT | | | WL | | | *p*-value Cohen's *d* | | |
| --- | --- | --- | --- | --- | --- | --- | --- | --- | --- | --- | --- |
| | | | N = 31 | | | N = 29 | | | Pre | Intervention period changes | Study period changes |
| | | | Pre | Post | FU | Pre | Post | FU | | | |
| CISS | T | Mean | 49.5 | 52.2 | 51.6 | 51.1 | 49.7 | 52.3 | 0.95 | 0.07 | 0.83 |
| | | SD | 10.2 | 10.3 | 9.6 | 12.9 | 14.1 | 15.6 | -0.02 | 0.53 | 0.06 |
| | E | Mean | 52.9 | 50.0 | 47.9 | 50.8 | 50.9 | 50.1 | 0.44 | 0.19 | 0.04* |
| | | SD | 11.3 | 11.0 | 11.8 | 10.1 | 10.4 | 10.7 | 0.21 | -0.37 | -0.59 |
| | A | Mean | 42.9 | 43.2 | 44.0 | 36.4 | 35.2 | 35.0 | 0.01* | 0.57 | 0.54 |
| | | SD | 10.1 | 9.3 | 12.3 | 10.8 | 11.0 | 9.1 | 0.77 | 0.16 | 0.17 |
| TAS20 | F1 | Mean | 22.9 | 21.3 | 21.0 | 23.1 | 23.2 | 23.6 | 0.59 | 0.20 | 0.14 |
| | | SD | 6.5 | 7.1 | 7.0 | 6.0 | 7.2 | 7.2 | -0.14 | -0.36 | -0.40 |
| | F2 | Mean | 18.4 | 17.4 | 17.0 | 19.7 | 20.2 | 19.3 | 0.08 | 0.04* | 0.42 |
| | | SD | 3.9 | 3.6 | 4.1 | 3.3 | 3.5 | 3.7 | -0.47 | -0.57 | -0.22 |
| | F3 | Mean | 21.5 | 19.7 | 20.5 | 20.4 | 19.7 | 20.5 | 0.52 | 0.23 | 0.44 |
| | | SD | 5.1 | 4.0 | 4.5 | 4.1 | 3.3 | 4.1 | 0.17 | -0.33 | -0.21 |
| | Total | Mean | 65.0 | 63.7 | 62.1 | 66.7 | 67.3 | 67.4 | 0.24 | 0.30 | 0.19 |
| | | SD | 8.4 | 10.4 | 11.8 | 9.0 | 9.3 | 10.2 | -0.31 | -0.29 | -0.36 |
| MPMR | | Mean | 69.7 | 77.3 | 79.9 | 66.3 | 70.9 | 72.2 | 0.36 | 0.36 | 0.30 |
| | | SD | 14.5 | 12.5 | 9.7 | 15.8 | 16.2 | 15.4 | 0.24 | 0.25 | 0.28 |
| ASD-Q | Knowledge | Mean | 7.8 | 8.6 | 8.4 | 7.6 | 8.0 | 8.2 | 0.60 | 0.42 | 0.88 |
| | | SD | 1.8 | 1.6 | 1.5 | 2.3 | 2.0 | 1.8 | 0.14 | 0.21 | 0.04 |
| | Attitude | Mean | 18.2 | 19.4 | 19.0 | 18.9 | 18.3 | 18.8 | 0.59 | 0.03* | 0.20 |
| | | SD | 3.2 | 3.7 | 3.1 | 3.6 | 4.7 | 3.8 | -0.14 | 0.59 | 0.34 |
| GAF | | Mean | 44.5 | 52.3 | 56.3 | 44.1 | 47.8 | 50.7 | 0.83 | 0.08 | 0.05 |
| | | SD | 8.8 | 8.3 | 8.4 | 7.9 | 8.0 | 6.8 | 0.06 | 0.48 | 0.53 |
| QOL | | Mean | 3.0 | 3.2 | 3.0 | 2.7 | 2.7 | 2.7 | 0.01* | 0.53 | 0.69 |
| | | SD | 0.6 | 0.6 | 0.7 | 0.6 | 0.8 | 0.7 | 0.50 | 0.17 | -0.11 |
| LSAS | Fear | Mean | 31.1 | 26.8 | 24.1 | 33.9 | 33.7 | 32.8 | 0.33 | 0.15 | 0.06 |
| | | SD | 15.0 | 11.5 | 14.2 | 13.8 | 12.0 | 12.8 | -0.26 | -0.39 | -0.51 |
| | Avoidance | Mean | 23.2 | 21.0 | 19.5 | 26.0 | 29.3 | 27.9 | 0.29 | 0.06 | 0.07 |
| | | SD | 13.8 | 11.6 | 14.0 | 11.8 | 15.5 | 14.1 | -0.27 | -0.52 | -0.50 |
| STAI | State | Mean | 47.6 | 46.1 | 49.3 | 54.8 | 53.6 | 52.8 | 0.06 | 0.90 | 0.38 |
| | | SD | 14.2 | 11.0 | 14.8 | 11.9 | 12.4 | 14.0 | -0.50 | -0.04 | 0.24 |
| | Trait | Mean | 57.4 | 53.5 | 53.8 | 62.2 | 59.9 | 57.1 | 0.10 | 0.49 | 0.80 |
| | | SD | 12.6 | 12.8 | 13.2 | 9.4 | 11.6 | 10.9 | -0.43 | -0.19 | 0.07 |
| SPAI | Social phobia | Mean | 114.0 | 107.9 | 100.1 | 132.2 | 128.4 | 127.8 | 0.03* | 0.68 | 0.16 |
| | | SD | 43.3 | 37.9 | 44.6 | 29.7 | 36.6 | 38.3 | -0.59 | -0.11 | -0.38 |
| | Agoraphobia | Mean | 26.3 | 24.5 | 23.0 | 27.4 | 27.4 | 26.1 | 0.79 | 0.52 | 0.32 |
| | | SD | 15.1 | 14.0 | 15.7 | 14.0 | 17.6 | 19.0 | -0.07 | -0.18 | -0.27 |
| CES-D | | Mean | 23.6 | 19.3 | 22.8 | 28.9 | 26.6 | 25.1 | 0.11 | 0.42 | 0.45 |
| | | SD | 12.9 | 10.7 | 10.8 | 12.9 | 14.7 | 13.8 | -0.42 | -0.23 | 0.21 |

Abbreviations: CBT, cognitive-behavioral therapy; WL, waitlist; Pre, pre-intervention; Post, post-intervention; FU, follow-up; CISS, the Coping Inventory for Stressful Situations; T, Task-oriented; E, Emotion-oriented; A, Avoidance-oriented; ASD-Q, autism spectrum disorder questionnaire; TAS20, 20-item Toronto Alexithymia Scale; F1, Difficulty Identifying Feelings Subscale; F2, Difficulty Describing Feelings Subscale; F3, Externally Oriented Thinking Subscale; MPMR, the Motion Picture Mind-Reading task; GAF, Global Assessment of Functioning; QOL, 26-item World Health Organization Quality of Life scale; LSAS, Liebowitz Social Anxiety Scale; SPAI, Social Phobia and Anxiety Inventory; STAI, State-Trait Anxiety Inventory; CES-D, Center for Epidemiological Studies Depression Scale.

Significance:

* = $p < 0.05$.

## Secondary outcomes

Results for all outcome measures are presented in Table 3. We found no significant differences in pre-intervention assessments in the GAF, QOL, LSAS, STAI, SPAI, and CES-D scores between the CBT and WL groups, except for the QOL and SPAI social phobia scores.

In the study period changes in secondary outcomes, independent $t$-test showed significant between-group differences in the CISS-E scores ($t$ = -2.14, $p$ = 0.04, $d$ = -0.59). With regard to the CISS-T, CISS-A, ASD-Q knowledge and attitude scores, TAS20-F1, TAS20-F2, TAS20-F3, TAS20 total scores, and MPMR scores, we found no significant between-group differences in study period changes.

For other secondary outcomes, including the GAF, QOL, LSAS, STAI, SPAI, and CES-D scores, we found no significant between-group differences in either intervention period changes or study period changes.

Additionally, we conducted a repeated measure analysis of variance to explore time changes in the three outcomes over time, which showed significant results in the $t$-tests (i.e., TAS20-F2, ASD-Q attitude, and CISS-E). Specifically, treatment group (i.e., the CBT vs WL control group) was considered as the between-subjects factor, while time point (i.e., pre-intervention, post-intervention, and FU) was considered as the within-subjects factor. With respect to the TAS20-F2 scores, a significant effect of treatment group was found ($F$ = 5.81, $p$ = 0.019), whereas we did not observe a main effect of time ($F$ = 1.33, $p$ = 0.27) or a group-by-time interaction ($F$ = 1.57, $p$ = 0.22). With regards to the ASD-Q attitude scores, a group-by-time interaction and a main effect of treatment group or time were not seen ($F$ = 3.03, $p$ = 0.05; $F$ = 0.05, $p$ = 0.82; $F$ = 0.65, $p$ = 0.52, respectively). Finally, with respect to the CISS-E, a significant effect of time was found ($F$ = 4.69, $p$ = 0.02), whereas we did not observe a group-by-time interaction ($F$ = 3.23, $p$ = 0.05) or a main effect of treatment group ($F$ = 0.03, $p$ = 0.86).

## Discussion

We found that group-based CBT on emotion regulation for autistic adults improved participants' attitude towards ASD, their difficulties in describing feelings, and their maladaptive emotion regulation strategies. Moreover, we demonstrated an acceptable adherence to a current CBT intervention program.

Participants in the two groups significantly differed in their intervention period score change in the TAS20 difficulties in describing feelings scale. However, the post-intervention total TAS20 score in the CBT group remained high, and participants in this group were still classified as having a severe impairment (a score of 61–100) [55]. Therefore, the CBT program had a measurable effect on describing feelings but was still insufficient to alleviate alexithymia generally.

Previous studies have reported that psychoeducation helped autistic children [56] and families with autistic adults [57] develop more positive outlooks towards ASD. Our results were consistent with those studies; however, while positive changes in attitude towards ASD were observed immediately after the intervention, they were not sustained 12 weeks later, indicating that the clinical utility of our CBT program on positive attitude towards ASD remained limited.

The reduction in maladaptive strategies suggests that the CBT program improved emotion regulation. Furthermore, improvements at the FU assessment might reflect continued post-intervention success in emotion regulation in daily life. However, the present data were insufficient to clarify the mechanisms underlying the delay in this effect on emotion regulation. Emotion regulation studies in adults have demonstrated that autistic adults use less task-oriented strategies but more emotion-oriented strategies than TD adults [9, 14]. Although the current

CBT program seems meaningful for reducing maladaptive strategies such as emotion-oriented strategies, future studies should aim to increase adaptive strategies, such as task-oriented strategies. We detected no significant changes in the MPMR scores over the intervention period. We speculated that alexithymia in ASD is related to deficits in the "theory of mind" based on previous studies [14, 22] and hypothesized that the ASD-focused CBT program would improve the ability to comprehend others' emotions through the identification of one's own emotions, or vice versa. However, our present findings do not support this hypothesis. Since the correct response rate increased in both groups, task repetition might have improved scores.

The secondary outcomes, which were used to assess the effects of the CBT program on adaptation, anxiety symptoms, and depression symptoms, did not significantly differ between the CBT and WL groups. Emotion regulation has been used to predict the symptoms of anxiety and depression in ASD [12, 13], and these emotional symptoms can be used to predict the adaptation of autistic individuals [5, 6]. However, the present CBT program, which showed effects on emotion regulation, did not reduce the emotional symptoms or improve adaptations. These negative results might be attributable to the relatively short-term intervention and study periods. Hence, future studies should elucidate the long-term effects of this intervention program, including the effects on adaptation and emotional symptoms and the improvements in emotion regulation brought on by the CBT program.

The initial power calculation proved to be rather optimistic, given that the observed effect sizes for the primary outcomes were much lower than expected (TAS20-F2 scores: $d = -0.57$; ASD-Q attitude scores: $d = 0.59$). Therefore, further differences between the two groups may not have been detected in the present study. The sample size for a larger randomized clinical trial is calculated on the basis of the effect sizes of the changes in TAS20-F2 scores ($d = -0.57$) and ASD-Q attitude scores ($d = 0.59$) during the intervention period. To detect a mean difference with a two-sided significance level of 5% and power of 80% with equal allocation to the two arms, this study would require between 47 and 50 participants in each arm of the trial. Although a larger sample could have been used to increase power of the between-group effects, the sample size was defined prior to the start of the study based on the initial power calculation and this could not be altered retrospectively.

This study has several limitations. First, the use of a waitlist control group limits the interpretation of the findings; therefore, it remains unclear which aspects of the CBT program are associated with changes in emotion regulation. Future studies should use more active control conditions (e.g., a group that does not include psychoeducation about ASD and emotion regulation) to delineate the active ingredients of the CBT program more precisely. Second, the assessments of improved outcomes were measured using self-report methods. Therefore, the placebo effect may have affected the results. Future studies should assess these components using blinded assessments.

In conclusion, our study provides preliminary evidence of the efficacy of the group-based CBT program on emotion regulation in autistic adults, thereby supporting a further evaluation of its effectiveness in a larger randomized clinical trial. However, the modest and inconsistent effects underscore the importance of continued efforts to improve the CBT program beyond current standards.

## Supporting information

**S1 File. CONSORT checklist.**
(PDF)

**S2 File. ASD knowledge and attitude quiz.**
(PDF)

**S3 File. Study protocol.**
(PDF)

**S4 File. Study protocol Japanese version.**
(PDF)

**S5 File. Minimal underlying data set.**
(XLSX)

## Author Contributions

**Conceptualization:** Miho Kuroda, Yuki Kawakubo, Hidenori Yamasue, Hitoshi Kuwabara.

**Data curation:** Miho Kuroda, Yuki Kawakubo.

**Formal analysis:** Miho Kuroda, Yuki Kawakubo.

**Funding acquisition:** Miho Kuroda.

**Investigation:** Miho Kuroda, Yuki Kawakubo, Yoko Kamio, Hitoshi Kuwabara.

**Methodology:** Miho Kuroda, Yuki Kawakubo, Hidenori Yamasue, Hitoshi Kuwabara.

**Project administration:** Miho Kuroda, Yuki Kawakubo, Yoko Kamio, Hidenori Yamasue, Toshiaki Kono, Maiko Nonaka, Natsumi Matsuda, Muneko Kataoka, Hitoshi Kuwabara.

**Resources:** Akio Wakabayashi, Kazuhito Yokoyama.

**Supervision:** Yukiko Kano.

**Writing – original draft:** Miho Kuroda, Yuki Kawakubo, Hitoshi Kuwabara.

**Writing – review & editing:** Miho Kuroda, Hitoshi Kuwabara.

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
