## [Decision Letter · Decision Letter 0]

23 Aug 2021

PONE-D-21-17599

Preliminary efficacy of cognitive behavioral therapy on emotion regulation in adults with autism spectrum disorder: a pilot randomized waitlist-controlled study

PLOS ONE

Dear Dr. Kuwabara,

Thank you for submitting your manuscript to PLOS ONE. After careful consideration, we feel that it has merit but does not fully meet PLOS ONE’s publication criteria as it currently stands. Therefore, we invite you to submit a revised version of the manuscript that addresses the points raised during the review process.

We look forward to receiving your revised manuscript.

Kind regards,

Valsamma Eapen, MBBS, PhD, FRCPsych, FRANZCP

Academic Editor

PLOS ONE

Journal Requirements:

This study was supported by an Intramural Research Grant (23–1) from the Neurological and Psychiatric Disorders program of the National Center of Neurology and Psychiatry and Grants-in-Aid (No. 23119706 and No. 22531078).

Reviewers' comments:

Reviewer's Responses to Questions

**Comments to the Author**

1. Is the manuscript technically sound, and do the data support the conclusions?

Reviewer #1: Partly

Reviewer #2: No

2. Has the statistical analysis been performed appropriately and rigorously? 

Reviewer #1: No

Reviewer #2: Yes

3. Have the authors made all data underlying the findings in their manuscript fully available?

Reviewer #1: Yes

Reviewer #2: Yes

4. Is the manuscript presented in an intelligible fashion and written in standard English?

Reviewer #1: Yes

Reviewer #2: Yes

5. Review Comments to the Author

Reviewer #1: The manuscript entitled ‘Preliminary efficacy of cognitive behavioral therapy on emotion regulation in adults with

autism spectrum disorder: a pilot randomized waitlist-controlled study’ with the aim to explore the preliminary efficacy of CBT on emotion regulation in adults with ASD.

Comments

Materials and Methods

Line 12, 15, 135-137, the time period Week 0-4, Week 4-8, Week 20 could be used to indicate the time period of assessment at pre, post and follow up.

Statistical analyses

Line 148, one or 2 tailed test to be stated.

If inferential statistics/p value is used/determined for the pilot study, at least sample size calculation could be added even though the study was an exploration. Likewise with repeated measure statistical test and multiple comparison correction.

The strength/findings of Cohen's d to be highlighted/discussed.

Results

Table 2, the decimal points for the p values to be standardized. Nonetheless, based on CONSORT statement, all statistical analyses for baseline comparison to be avoided.

Missing data to be stated if any.

Line 320-330, table to be cited.

Figure 1, the period of assessment to be incorporated in

List of references to follow PLoS ONE format.

Reviewer #2: Thank you for the opportunity to review this paper which conducted a randomised clinical trial exploring a CBT program intervention in autistic adults. I feel that some very good work was done, though overall that the largely non-significant results do not support the framing of the discussion of conclusions. My specific comments as follows:

Stylistic issue, I do prefer identity first language and note the general shift in many academic autism literature to identity first given the stated preferences of the autistic adult community. Though this is a decision for the authors and editor.

The statement “Moreover, the implementation of prominent emotion regulation strategy patterns in ASD has been associated with mental disorders” is unclear to me, I suggest it needs more explanation.

I would suggest the reference to theory of mind in the introduction is overly simplistic, and does not capture the controversy and latest research regarding the claim of theory of mind impairment in autistic adults.

The inclusion criteria of “All participants needed to have an awareness of their lack of emotional self-awareness” seems vague to me, how was this assessed?

I feel the ASD Quiz might be better named along the lines of an autism knowledge and attitude quiz to make it easier for the reader to follow.

I would suggest a paragraph break before describing the TAS20 would improve readability.

I feel the discussion also needs to open describing the general lack of change across the majority of measures. Overall, I think that, although more work and research is needed, the discussion needs to be forthcoming in suggesting that there may only minimal value in CBT for autistic adults based on findings in this study. I suggest perhaps the authors engage with grey literature from autistic adults who are critical of CBT approaches to help contextualise why these findings may occur. I am not sure entirely of the conclusion being made given the generally non-significant results.

6. PLOS authors have the option to publish the peer review history of their article (what does this mean?). If published, this will include your full peer review and any attached files.

Reviewer #1: No

Reviewer #2: No

---

## [Author Response · Author response to Decision Letter 0]

30 Sep 2021

We would like to thank the Reviewers for their insightful comments and suggestions, which we believe have helped us improve our manuscript and provide a more balanced account of our research. We have carefully reviewed our manuscript according to their comments and made the necessary changes, which have been tracked in the revised manuscript to facilitate the review process. Please find below our point-by-point responses to all comments.

Reviewer #1: 

Line 12, 15, 135-137, the time period Week 0-4, Week 4-8, Week 20 could be used to indicate the time period of assessment at pre, post and follow up.

Response: We would like to thank you for your valuable suggestion, we have now revised our manuscript accordingly. (Page 4, Lines: 12-13, 16; Page 11, Lines: 141-142, 143)

Line 148, one or 2 tailed test to be stated.

Response: We would like to thank you for your comment and apologize for the lack of clarity, we have now revised our manuscript accordingly. (Page 12, Line 154)

If inferential statistics/p value is used/determined for the pilot study, at least sample size calculation could be added even though the study was an exploration. Likewise with repeated measure statistical test and multiple comparison correction.

Response: We would like to thank you for your insightful suggestion. We confirm that we have now added the requested information to our revised manuscript as follows: 

“The sample size for a larger randomized clinical trial is calculated on the basis of the effect sizes of the changes in TAS20-F2 scores (d = -0.57) and ASD-Q attitude scores (d = 0.59) during the intervention period. To detect a mean difference with a two-sided significance level of 5% and power of 80% with equal allocation to the two arms, this study would require 47-50 participants in each arm of the trial.” (Page 29, Lines: 412-416)

“Additionally, we conducted a repeated measure analysis of variance to explore the changes in the three outcomes over time, which showed significant results in the t-tests (i.e., TAS20-F2, ASD-Q attitude, and CISS-E). Specifically, treatment group (i.e., the CBT vs WL control group) was considered as the between-subjects factor, while time point (i.e., pre-intervention, post-intervention, and follow-up) as the within-subjects factor. With respect to the TAS20-F2 scores, a significant effect of treatment group was found (F = 5.81, p = 0.19), whereas we did not observe a main effect of time (F = 1.33, p = 0.27) or a group-by-time interaction (F = 1.57, p = 0.22). With regards to the ASD-Q attitude scores, a group-by-time interaction and a main effect of treatment group or time were not seen (F = 3.03, p = 0.05; F = 0.05, p = 0.82; F = 0.65, p = 0.52, respectively). Finally, with respect to the CISS-E, a significant effect of time was found (F = 4.69, p = 0.02), whereas we did not observe a group-by-time interaction (F=3.23, p = 0.05) or a main effect of treatment group (F = 0.03, p = 0.86).” (Page 25, Lines: 336-348)

“This being said, these findings did not withstand the Bonferroni correction for multiple comparisons for the 10 primary outcome measures (p < 0.05/10).” (Page 22, Lines: 303-304)

The strength/findings of Cohen’s d to be highlighted/discussed.

Response: We would like to thank you for your insightful comment and apologize for the lack of information. We confirm that we have now added the following sentences to the “Discussion” section of our revised manuscript to increase clarity:

“The initial power calculation proved to be rather optimistic, given that the observed effect sizes for the primary outcomes were much lower than expected (TAS20-F2 scores; d = -0.57 and ASD-Q attitude scores; d = 0.59). Therefore, further differences between the two groups may have not been detected in the present study. Although a larger sample could have been used to increase the power of the between-group effects, the sample size was defined prior to the start of the study based on the initial power calculation and this could not be altered in retrospect.” (Page 28, Lines: 394-400)

Table 2, the decimal points for the p values to be standardized. Nonetheless, based on CONSORT statement, all statistical analyses for baseline comparison to be avoided.

Response: We would like to thank you for your comment. We have now deleted all statistical analyses for baseline comparison from our revised manuscript, as suggested.

Missing data to be stated if any.

Response: We would like to thank you for your valuable comment. We have now states the above in the annotation of Table 2. (Page 21, Line 289)

Line 320-330, table to be cited.

Response: We would like to thank you for your valuable comment. We are pleased to inform you that our manuscript has now been revised accordingly. (Page 24, Line 324)

Figure 1, the period of assessment to be incorporated in

Response: We would like to thank you for your comment. We have now revised Figure 1 according to your suggestion.

List of references to follow PLoS ONE format.

Response: We would like to thank you for your valuable comment and apologize for the mistake. We have now revised the List of References according to the PLOS ONE ‘s style requirements.

Reviewer #2: 

Stylistic issue, I do prefer identity first language and note the general shift in many academic autism literature to identity first given the stated preferences of the autistic adult community. Though this is a decision for the authors and editor.

Response: We would like to thank you for your insightful comment. We confirm that we have now changed the description to identity first language throughout our revised manuscript to address your comment.

The statement “Moreover, the implementation of prominent emotion regulation strategy patterns in ASD has been associated with mental disorders” is unclear to me, I suggest it needs more explanation.

Response: We would like to thank you for your insightful comment and apologize for the lack of clarity. We have now noticed that the meaning of the mentioned statement overlapped with the previous sentence (Page 5, Line 33), ”Studies on emotion regulation in both children and ASD adults have primarily focused on differences in the implementation of particular emotion regulation strategy, in addition to the relationship between particular emotion regulation and a range of outcomes, including mental health and social functioning”. Therefore we have now deleted the mentioned statement from our revised manuscript to increase accuracy.

I would suggest the reference to theory of mind in the introduction is overly simplistic, and does not capture the controversy and latest research regarding the claim of theory of mind impairment in autistic adults.

Response: We would like to thank you for your valuable suggestion. We have now added the following sentences to the “Introduction” section of our revised manuscript to increase clarity:

“Theory of mind, i.e., the ability to attribute mental states to others to make sense of their behavior, has been previously suggested to be atypical in ASD [18, 19]. However, it remains unknown whether a universal pattern of cognitive impairment in ASD exists and whether multiple cognitive impairments are needed to explain its full range of behavioral symptoms [20]. Social cognition clearly encompasses a range of processes, including, but not limited to, theory of mind and emotion processing, which appear to be distinct but interdependent [21].” (Page 6, Line 54; Page 7, Lines: 55-61)

As a consequence, the following references were added to our revised Reference List:

18. Cantio C, Jepsen JR, Madsen GF, Bilenberg N, White SJ. Exploring 'The autisms' at a cognitive level. Autism Res. 2016;9(12):1328-39. doi: 10.1002/aur.1630.

19. Happe F, Cook JL, Bird G. The Structure of Social Cognition: In(ter)dependence of Sociocognitive Processes. Annu Rev Psychol. 2017;68:243-67. doi: 10.1146/annurev-psych-010416-044046.

20. Livingston LA, Happe F. Conceptualising compensation in neurodevelopmental disorders: Reflections from autism spectrum disorder. Neurosci Biobehav Rev. 2017;80:729-42. doi: 10.1016/j.neubiorev.2017.06.005.

21. Brewer R, Happe F, Cook R, Bird G. Commentary on "Autism, oxytocin and interoception": Alexithymia, not Autism Spectrum Disorders, is the consequence of interoceptive failure. Neurosci Biobehav Rev. 2015;56:348-53. doi: 10.1016/j.neubiorev.2015.07.006.

The inclusion criteria of “All participants needed to have an awareness of their lack of emotional self-awareness” seems vague to me, how was this assessed?

Response: We would like to thank you for your valuable comment and apologize for the lack of clarity. Participants’ awareness was assessed through direct interview with a trained psychologists. We have now added the following sentences to the “Materials and methods” section of our revised manuscript to increase understanding:

“Participants’ awareness of their lack of emotional self-awareness and poor comprehension of others’ emotions was confirmed through a direct interview with a PhD-qualified psychologist.” (Page 11, Lines: 128-130)

I feel the ASD Quiz might be better named along the lines of an autism knowledge and attitude quiz to make it easier for the reader to follow.

Response: We would like to thank you for your insightful suggestion. We have now revised the title of the ASD Quiz to “ASD knowledge and attitude quiz.” (Page 15, Line 202; Page 30, Line 423, Supporting information S2 File)

I would suggest a paragraph break before describing the TAS20 would improve readability.

Response: We would like to thank you for your valuable comment. We have now revised our manuscript accordingly. (Page 16, Line 215).

I feel the discussion also needs to open describing the general lack of change across the majority of measures. Overall, I think that, although more work and research is needed, the discussion needs to be forthcoming in suggesting that there may only minimal value in CBT for autistic adults based on findings in this study. I suggest perhaps the authors engage with grey literature from autistic adults who are critical of CBT approaches to help contextualise why these findings may occur. I am not sure entirely of the conclusion being made given the generally non-significant results.

Response: We would like to thank you for your insightful comment. We have now added the following explanation of the limited significance in the differences between groups to the “Discussion” section of our revised manuscript to increase accuracy:

“The initial power calculation proved to be rather optimistic, given that the observed effect sizes for the primary outcomes were much lower than expected (TAS20-F2 scores; d = -0.57 and ASD-Q attitude scores; d = 0.59). Therefore, further differences between the two groups may have not been detected in the present study. Although a larger sample could have been used to increase the power of the between-group effects, the sample size was defined prior to the start of the study based on the initial power calculation and could not be altered in retrospect.” (Page 28, Lines: 394-400)

---

## [Decision Letter · Decision Letter 1]

4 Jul 2022

PONE-D-21-17599R1Preliminary efficacy of cognitive behavioral therapy on emotion regulation in autism spectrum disorder adults: a pilot randomized waitlist-controlled studyPLOS ONE

Dear Dr. Kuwabara,

Thank you for submitting your manuscript to PLOS ONE. After careful consideration, we feel that it has merit but does not fully meet PLOS ONE’s publication criteria as it currently stands. Therefore, we invite you to submit a revised version of the manuscript that addresses the points raised during the review process.

 While the changes require a major revision, I believe they can be reasonably addressed.

We look forward to receiving your revised manuscript.

Kind regards,

Tarek K Rajji

Academic Editor

PLOS ONE

Reviewers' comments:

Reviewer's Responses to Questions

**Comments to the Author**

1. If the authors have adequately addressed your comments raised in a previous round of review and you feel that this manuscript is now acceptable for publication, you may indicate that here to bypass the “Comments to the Author” section, enter your conflict of interest statement in the “Confidential to Editor” section, and submit your "Accept" recommendation.

Reviewer #1: (No Response)

Reviewer #2: (No Response)

2. Is the manuscript technically sound, and do the data support the conclusions?

Reviewer #1: Yes

Reviewer #2: Partly

3. Has the statistical analysis been performed appropriately and rigorously? 

Reviewer #1: No

Reviewer #2: No

4. Have the authors made all data underlying the findings in their manuscript fully available?

Reviewer #1: Yes

Reviewer #2: (No Response)

5. Is the manuscript presented in an intelligible fashion and written in standard English?

Reviewer #1: Yes

Reviewer #2: Yes

6. Review Comments to the Author

Reviewer #1: The authors have put in great effort to address the comments.

Minor comment(s)

Line 335, Repeated measures ANOVA comes with pairwise/multiple comparison for post hoc test. If based on repeated measures ANOVA approach, the use of t test to be clarified in the text.

Reviewer #2: Thank you for the opportunity to again review this manuscript. I still feel the discussion and abstract needs to be more forthcoming describing the general lack of change across the majority of measures. Overall, I think that, although more work and research is needed, the discussion needs to be cautious in suggesting that there may only minimal value in group CBT for autistic adults based on findings in this study.

An additional outstanding concern is the use of language. When I refer to identity first, I mean using the term “autistic adult” not “ASD adult” or “ASD individual”. Alternatively neutral language, which you could perhaps use at the start of the introduction, would be “Adults on the autism spectrum”. Also I think it’s better to refer to autism generally after an initial first mention of ASD.

Just checking if it should be an ‘or’ not an ‘and’ in the list of tools used in diagnostic confirmation?

In the analyses section you say no adjustment was made for multiple comparisons, then in results a Bonferroni correction is described?

Significance is unclear to me when it is reported “With 356 respect to the TAS20-F2 scores, a significant effect of treatment group was found (F = 5.81, p = 0.19)” given the p value?

7. PLOS authors have the option to publish the peer review history of their article (what does this mean?). If published, this will include your full peer review and any attached files.

Reviewer #1: No

Reviewer #2: No

---

## [Author Response · Author response to Decision Letter 1]

5 Aug 2022

We would like to thank the reviewers for their insightful comments and suggestions, which have helped us improve our manuscript and provide a more balanced account of our research. We have carefully reviewed our manuscript in light of their comments and made the necessary changes, which are shown in red font in the revised manuscript to facilitate the review process. Please find below our point-by-point responses to their comments.

Reviewer #1

Line 335, Repeated measures ANOVA comes with pairwise/multiple comparison for post hoc test. If based on repeated measures ANOVA approach, the use of t test to be clarified in the text.

Response: Thank you for your valuable suggestion. We have revised our manuscript accordingly (page 21, line 305; page 24, line 333).

Reviewer #2

I still feel the discussion and abstract needs to be more forthcoming describing the general lack of change across the majority of measures. Overall, I think that, although more work and research is needed, the discussion needs to be cautious in suggesting that there may only minimal value in group CBT for autistic adults based on findings in this study.

Response: Thank you for your insightful comment. We have revised the discussion section (page 28, lines 405-407), and added the following explanation of the general lack of differences between groups to both the discussion section and the abstract of our revised manuscript to be more forthcoming:

“However, the modest and inconsistent effects underscore the importance of continued efforts to improve the CBT program program beyond current standards” (page 4, lines 23-24; pages 28-29, lines 421-423).

An additional outstanding concern is the use of language. When I refer to identity first, I mean using the term “autistic adult” not “ASD adult” or “ASD individual”. Alternatively neutral language, which you could perhaps use at the start of the introduction, would be “Adults on the autism spectrum”. Also I think it’s better to refer to autism generally after an initial first mention of ASD.

Response: Thank you for this valuable comment. In response to your comment, we changed the description to “autistic” instead of “ASD” throughout our revised manuscript.

Just checking if it should be an ‘or’ not an ‘and’ in the list of tools used in diagnostic confirmation?

Response: We apologize for the mistake. It should be “or”. We have revised the manuscript accordingly (page 9, line 126).

In the analyses section you say no adjustment was made for multiple comparisons, then in results a Bonferroni correction is described?

Response: We appreciate your comment. We deleted the following sentence from the analyses section: “We did not conduct a correction for the multiple comparisons due to the exploratory nature of this study.”

Significance is unclear to me when it is reported “With 356 respect to the TAS20-F2 scores, a significant effect of treatment group was found (F = 5.81, p = 0.19)” given the p value?

Response: We apologize for the mistake. It should be “p = 0.019”. We have revised the manuscript accordingly (page 24, line 347).

---

## [Decision Letter · Decision Letter 2]

6 Sep 2022

PONE-D-21-17599R2Preliminary efficacy of cognitive-behavioral therapy on emotion regulation in adults with autism spectrum disorder: a pilot randomized waitlist-controlled studyPLOS ONE

Dear Dr. Kuwabara,

Thank you for submitting your manuscript to PLOS ONE. After careful consideration, we feel that it has merit but does not fully meet PLOS ONE’s publication criteria as it currently stands. Therefore, we invite you to submit a revised version of the manuscript that addresses the points raised during the review process.

We look forward to receiving your revised manuscript.

Kind regards,

Tarek K Rajji

Academic Editor

PLOS ONE

Journal Requirements:

Reviewers' comments:

Reviewer's Responses to Questions

**Comments to the Author**

1. If the authors have adequately addressed your comments raised in a previous round of review and you feel that this manuscript is now acceptable for publication, you may indicate that here to bypass the “Comments to the Author” section, enter your conflict of interest statement in the “Confidential to Editor” section, and submit your "Accept" recommendation.

Reviewer #1: All comments have been addressed

Reviewer #2: (No Response)

2. Is the manuscript technically sound, and do the data support the conclusions?

Reviewer #1: (No Response)

Reviewer #2: Yes

3. Has the statistical analysis been performed appropriately and rigorously? 

Reviewer #1: (No Response)

Reviewer #2: Yes

4. Have the authors made all data underlying the findings in their manuscript fully available?

Reviewer #1: (No Response)

Reviewer #2: Yes

5. Is the manuscript presented in an intelligible fashion and written in standard English?

Reviewer #1: (No Response)

Reviewer #2: Yes

6. Review Comments to the Author

Reviewer #1: (No Response)

Reviewer #2: Thank you for addressing previous suggestions. I noticed - "it remains unclear which aspects of the CBT program are associated with changes in ASD" - I don't believe ASD changes and feel this sentence should be revised.

7. PLOS authors have the option to publish the peer review history of their article (what does this mean?). If published, this will include your full peer review and any attached files.

Reviewer #1: No

Reviewer #2: No

---

## [Author Response · Author response to Decision Letter 2]

7 Sep 2022

Reviewer #2

Thank you for addressing previous suggestions. I noticed - "it remains unclear which aspects of the CBT program are associated with changes in ASD" - I don't believe ASD changes and feel this sentence should be revised.

Response: We apologize for the mistake. It should be “emotion regulation”. We have revised the manuscript accordingly (page 28, line 413).

---

## [Editor Report · Decision Letter 3]

27 Oct 2022

Preliminary efficacy of cognitive-behavioral therapy on emotion regulation in adults with autism spectrum disorder: a pilot randomized waitlist-controlled study

PONE-D-21-17599R3

Dear Dr. Kuwabara,

We’re pleased to inform you that your manuscript has been judged scientifically suitable for publication and will be formally accepted for publication once it meets all outstanding technical requirements.

Kind regards,

Tarek K Rajji

Academic Editor

PLOS ONE
---

## [Editor Report · Acceptance letter]

15 Nov 2022

PONE-D-21-17599R3 

Preliminary efficacy of cognitive-behavioral therapy on emotion regulation in adults with autism spectrum disorder: a pilot randomized waitlist-controlled study 

Dear Dr. Kuwabara:

I'm pleased to inform you that your manuscript has been deemed suitable for publication in PLOS ONE. Congratulations! Your manuscript is now with our production department. 

Kind regards, 

on behalf of

Dr. Tarek K Rajji 

Academic Editor

PLOS ONE